# Effect of chicken manure and chemical fertilizer on the yield and qualities of white mugwort at dissimilar harvesting times

Ornprapa Thepsilvisut[1]*, Preuk Chutimanukul[1], Sudathip Sae-Tan[2], Hiroshi Ehara[3]

1 Department of Agricultural Technology, Faculty of Science and Technology, Thammasat University, Khlong Nueng, Pathum Thani, Thailand, 2 Department of Food Science and Technology, Faculty of Agro-Industry, Kasetsart University, Bangkok, Thailand, 3 International Center for Research and Education in Agriculture, Nagoya University, Nagoya, Japan

* ornprapa@hotmail.com, ornprapa@staff.tu.ac.th

**Data Availability Statement:** All relevant data are within the paper.

## Abstract

One of the key components that affects soil productiveness, plant growth, and crop quality is fertilization. The effect of fertilizer, both organic and chemical, on the extremely acidic (pH 4.10) sandy loam soil chemical properties, yield, and quality of white mugwort grown were evaluated in this study. The field experiment arranged in a randomized complete block design, with four replications was conducted in Prachin Buri province, Thailand. There were six treatments, no fertilization (control), chemical fertilizer (25-7-7 + 46-0-0) applied at 187.50 (66.56 N:6.77 $P_2O_5$:6.77 $K_2O$ kg ha$^{-1}$) kg ha$^{-1}$ and applied at 375.00 (133.12 N:13.13 $P_2O_5$:13.13 $K_2O$ kg ha$^{-1}$) kg ha$^{-1}$, chicken manure applied at 3.36, 6.72, and 10.08 t ha$^{-1}$. After harvesting, application of chicken manure tended to increase soil organic matter compared to the control, however, the application of chemical fertilizer did not show the same effect. The fresh weight of white mugwort increased with the rise in both fertilizer levels. Chicken manure application at 10.08 t ha$^{-1}$ produced the highest fresh weight at all times. The level of accumulated nitrate was significantly greater when fertilizer rates increased. In addition, the total phenolic content of the white mugwort fertilized with chicken manure was higher than that fertilized with chemical fertilizer. However, there was no association between the DPPH radical scavenging capacity at harvesting times and different fertilization. Based on the results, chicken manure applied at 10.08 t ha$^{-1}$ gave the best yield and higher total phenolic content of white mugwort, which was probably due to the improved soil organic matter.

## Introduction

White mugwort (*Artemisia lactiflora* Wall. ex DC.) is one species of the genus *Artemisia* that belongs to the Compositae family and primarily exists in southeast Asia. In Thailand, the plant is known as "Jingjuchai", and its leaves are used in soups and aromatic herb tea. Previous research has examined white mugwort's phytochemical constituents, such as quercetin, rutin,

**Funding:** Thammasat University under the TU New Research Scholar fiscal year 2016. Role of Funder Statement: The funders had no role in study design, data collection and analysis, decision to publish, or preparation of the manuscript.

**Competing interests:** The authors have declared that no competing interests exist.

caffeic acid ethyl ester, and diacetylenic spiroacetal enol ethers [1]. Nakamura et al. [2] found spiroacetal enol ethers epoxide AL-1 inhibits superoxide ($O_2^-$) generation and prevents oxidative stress-related diseases, including cancers. In addition, it has been shown that the leaves of white mugwort are rich in total phenolic, ascorbic acid, and some essential minerals, such as iron and calcium [3].

Optimal fertilization is a principal factor for both plant growth and crops phytochemical quality. There are several reports on organic and chemical fertilizers variances that effect the nutritional and antioxidant contents in plant tissues [4, 5]. Ibrahim et al. [6] reported that the use of organic fertilizer (chicken manure) increases secondary metabolite production and antioxidant activity, such as gluthathione, total phenolics and DPPH radical scavenging of *Labisia pumila* Benth compared to using chemical fertilizers. To the best of our knowledge, no study has been published investigating the effects of distinct types and dosages of organic and chemical fertilizers on the growth and phytochemical composition of *A. lactiflora*. However, there have been several studies of *Artemisia annua*, a species that is congeneric with white mugwort, which reported that a higher application of nitrogen sources, both organic and chemical fertilizers, could increase the yield of artemisinin (secondary compounds for malaria therapy) in the leaves [7, 8]. A previous study by Ferreira [9] also found that nitrogen was the main element for promoting leaf biomass of Artemisia plants rather than phosphorus or potassium and thus increased the yield of artemisinin per unit area. An increase in foliage yield, leading to enhanced artemisinin production in *A. annua*, has been achieved by increasing the amount of chemical fertilizer [10]. However, many studies have shown a significantly higher concentration of nitrates in conventionally grown vegetables (using chemical fertilizer) such as lettuce [11], carrots [12], and watercress [13]. This may be attributed to the fact that inorganic nitrogen fertilizer easily dissolves and releases nitrogen, which is quickly absorbed by the plants, hence the higher nitrate accumulation. Since the elevated level of nitrate in vegetables has been linked to an increased risk of health problems [14], adequate fertilization management to ensure sufficient plant growth without nitrate accumulation in plant tissues has been gaining attention. A recent study showed that chicken manure application resulted in a highest yield and chlorophyll content in white mugwort when compared with the application of cow or bat manure. This suggests a potential for chicken manure in white mugwort cultivation [15]. Poultry manure was also reported to release nitrogen slowly, which resulted in significantly lower nitrate accumulation in plants [16]. Organic fertilizers, such as poultry manure, have been found to improve soils physical, chemical, and biological properties, including bulk density [17], organic carbon [18], and microbial biomass [19]. However, at present, there has not been any report about the effect of different dosages of chicken manure and chemical fertilizer on the yield and quality of white mugwort. Therefore, this study aimed to investigate the effects of chicken manure and chemical fertilizer on the yield and quality of white mugwort in terms of pigment concentrations, phenolic content, antioxidant activity (DPPH radical scavenging activity), and nitrate content, particularly in extremely acidic sandy loam soil.

## Materials and methods

### Experimental site and soil characteristics

The study field experiments were conducted in a farmer's field in Nadi district, Prachin Buri province (Latitude: 14.0847789, Longitude: 101.9875631) from April to September 2015. The soil of the experimental area was classified as Plinthaquic Paleustults (Buntharik soil series). The soil had a sandy loam texture and was very strongly to strongly acidic throughout the soil profile [20]. Soil samples were collected and placed into plastic bags for further analysis. Table 1 shows the properties of the chicken manure used in this experiment.

**Table 1. Properties of chicken manure used in the experiment.**

| Properties | Chicken manure |
|---|---|
| pH (1:5 $H_2O$) | 7.70 |
| EC (1:5 $H_2O$, dS $m^{-1}$) | 8.15 |
| Organic matter (g $kg^{-1}$)[a] | 559.30 |
| Total N (g $kg^{-1}$)[b] | 29.30 |
| Total P (g $kg^{-1}$)[c] | 43.20 |
| Total K (g $kg^{-1}$)[d] | 38.80 |
| Total Ca (g $kg^{-1}$)[d] | 47.50 |
| Total Mg (g $kg^{-1}$)[d] | 36.20 |

[a] Walkley and Black method.

[b] Kjeldahl method.

[c] Vanadomolybdate method.

[d] Wet digestion method.

## Experimental layout and management

A randomized complete block design with four replications was used to perform the study. The six treatments comprised non-fertilizer (Control), chemical fertilizers (complete fertilizer 25-7-7 grade + urea 46-0-0 grade) applied at rates of 187.50 (66.56:6.77:6.77 kg $ha^{-1}$ of N:$P_2O_5$:$K_2O$) and 375.00 (133.12:13.13:13.13 kg $ha^{-1}$ of N:$P_2O_5$:$K_2O$) kg $ha^{-1}$, and chicken manure applied at a rate of 3.36, 6.72 and 10.08 t $ha^{-1}$. The nutrient content of each fertilizer application is shown in Table 2.

Ten-centimeter clumps of white mugwort were transplanted into plastic bags (7.5 cm x 17.5 cm), containing rice husk charcoal. After 30 days of cultivation, sixty-four plants were transplanted into each of the six 4 $m^2$ plots with a plant to plant and row to row spacing of 25 cm. The doses of organic fertilizers "chicken manure" were applied at the time of transplanting, whereas the doses of chemical fertilizers were applied with complete fertilizer 25-7-7 grade at the time of transplanting, and with urea 46-0-0 grade at 14 days before harvest. Irrigation was provided equally to all plots by using a dripping irrigation system near the root at a rate of 6 L $h^{-1}$ for 15–20 min once a day. Neem extracts were sprayed to eradicate disease and insects when spreading and mechanical methods were utilized to remove the weeds. The plants were harvested three times at 42, 84, and 126 days after transplantation by cutting at 7-cm above the ground. The same type and amount of fertilizer for each treatment was applied after each harvesting.

**Table 2. Background plant nutrient contents when various fertilizers were applied at each time point.**

| Fertilizer treatment | Application rates (t $ha^{-1}$) | Proportion of primary elements in fertilizer applied for each treatment | | |
|---|---|---|---|---|
| | | N (kg $ha^{-1}$) | P (kg $ha^{-1}$) | K (kg $ha^{-1}$) |
| Control | 0 | 0 | 0 | 0 |
| CF 187.50 kg $ha^{-1}$ (25-7-7$_{93.75 kg ha^{-1}}$ + 46-0-0$_{93.75 kg ha^{-1}}$) | 0.19 | 66.56 | 6.77 | 6.77 |
| CF 375.00 kg $ha^{-1}$ (25-7-7$_{187.50 kg ha^{-1}}$ + 46-0-0$_{187.50 kg ha^{-1}}$) | 0.38 | 133.12 | 13.13 | 13.13 |
| CM 3.36 t $ha^{-1}$ | 3.36 | 66.56 | 145.15 | 130.37 |
| CM 6.72 t $ha^{-1}$ | 6.72 | 133.12 | 290.30 | 260.74 |
| CM 10.08 t $ha^{-1}$ | 10.08 | 199.68 | 435.46 | 391.10 |

## Data collection

**Soil chemical properties.** Soil samples were taken after three harvesting times and were air-dried, crushed and sieved through a 2 mm sieve for the analysis of specific soil chemical properties. The soil's pH and electrical conductivity (EC) were measured in water at a ratio of 1:1 and 1:5, respectively, and determined by using a pH-EC meter (SciberScanPC510, EUTEC, Singapore). The organic matter was measured by using the Walkley and Black method [21]. The total N content was determined by using the Kjeldahl method. The available P was extracted from the soil using the Bray II method and determined by using a spectrophotometer (UV-1280, Shimadzu, Japan) at 880 nm, while the available K was extracted from the soil using a 1M $NH_4OAc$ pH 7.0 and was measured by using a flame photometer (410, Sherwood Scientific Ltd., UK) [22].

**Plant growth and water content.** Ten plant samples were randomly selected from each treatment at 42, 84 and 126 days after transplanting. The fresh weight of the upper parts (7 cm height from the ground) was recorded. The fresh yield of the treated plants was calculated from the fresh weight of upper parts per plant that were cultivated under the 25 cm x 25 cm plant spacing, equating to 350,000 plant $ha^{-1}$. The samples were oven-dried at 50 ± 3˚C until a stable dry weight was achieved. For the determination of the dry weight and calculating the water content, the equation below was used:

$$\text{water content (\%)} = [(\text{fresh weight} - \text{dry weight})/\text{fresh weight}] \times 100$$

**Chlorophyll content.** After harvesting, the chlorophyll content of the white mugwort leaves was measured by using the method of Mackinney [23]. To determine the total chlorophyll content, a 0.28 $cm^2$ section was removed from the white mugwort leaf. Then, to extract the chlorophyll the section was soaked in 10 mL of 80% (v/v) acetone. A spectrophotometer was used to measure the supernatant at 645 and 663 nm to determine the total chlorophyl content and the content was expressed as content per unit surface area.

**Nitrate concentration.** The nitrate concentration was measured according to the applied method from AOAC [24], and the sample preparation method was adapted from the procedure reported by Anugoolprasert and Rithichai [25]. Shoot dry weight samples of 0.1 g were added to 10 mL of distilled water and incubated at 50 ± 3˚C for 24 h in a shaking water bath at 120 rpm. After that, 0.5 mL of the supernatant was used for colorimetric determination of nitrate by using the brucine–sulfanilic acid method at a wavelength of 410 nm by using a spectrophotometer. The concentration of the nitrate was calculated from the calibration curves of the standards and the sample weight was converted for reporting as g $kg^{-1}$ fresh weight.

**Phytochemical compounds.** Determination of total phenolic content followed the method of Anugoolprasert and Rithichai [25] and Liu et al. [26] by using Folin-Ciocalteau assay with some modifications. After weighing the homogenized sample (1 g of shoot fresh weight) using the ratio 1:10 w/v 10 mL of 70% (v/v) ethanol was added and stirred in a shaker at 150 rpm and 4˚C for 150 min. The extracted was separated from the residue by filtrating through a Whatman No. 1 filter paper. To find out the total phenolic content analysis, the extract (0.3 mL) was transferred to a test tube, 2.5 mL of 2 M Folin-Ciocalteu reagent and 2.0 mL of 7.5% (w/v) sodium carbonate were added and mixed. Subsequently, the mixture was incubated in the dark, at room temperature, for 150 min, afterward a spectrophotometer was used at a wavelength of 765 nm to measure the purity. The standard calibration curve was plotted by using gallic acid and total phenolic content was expressed as gallic acid equivalents in mg gallic acid per 1 g fresh weight. A modified method of the Wang et al. [27] and Jin et al. [28] method was carried out for DPPH assay. Succinctly, 0.1 mL of the extracted samples with

different concentrations in the microplate were mixed with 0.1 mL DPPH solution (0.1 mM in 95% ethanol). The samples were kept at room temperature, in the dark condition. After 30 min, a micro-plate reader at 517 nm was used to measure the absorbance. The standard used was Trolox and results were expressed as mg Trolox equivalent antioxidant capacity (TEAC) per g fresh weight.

## Statistical analysis

Analysis of variance in a randomized complete block design was conducted by using IBM SPSS Statistics, Version 26.0 software (IBM Corp., Armonk, NY, USA) was used to conduct analysis of variance in a randomized complete block design. The mean values of treatment difference were compared by using Duncan's multiple range tests, with the significance determined at $p < 0.05$.

## Results and discussion

### Soil chemical properties

The chemical properties of the soil samples before and after treatments are shown in Table 3. The soil pH at the experimental site was extremely acidic with a pH of 4.10 and the soil electrical conductivity (EC) was at 0.06 dS m$^{-1}$ (non-saline soil). Soil organic matter and total nitrogen level were low ($< 15$ g kg$^{-1}$ for organic matter and 3 g kg$^{-1}$ for total nitrogen), while the phosphorus level was very high ($> 45$ mg kg$^{-1}$) and the potassium level was moderate (61–90 mg kg$^{-1}$), according to the information from the Soil Survey Staff [29]. After the application of chicken manure and chemical fertilizer, the soil pH increased from 4.10 to 4.20–7.10 and the soil EC was found to be between 0.03–0.08 dS m$^{-1}$ at all three harvest times. Interestingly, the results showed that the higher rate of chicken manure application improved the soil chemical properties by increasing the organic matter, total N, available P, and available K at all three harvest times, these effects were not found in the chemical fertilizer soil. Chicken manure can slowly supply the plant nutrients and also has a high organic matter content (559.30 g kg$^{-1}$) which can improve soil's physical properties as reported by Lima et al. [17]. The soil applied with both fertilizers had available P between 122 to 274 mg kg$^{-1}$ and available K between 48 to 228 mg kg$^{-1}$. Interestingly, the higher dosage of chicken manure resulted in higher available P and K compared with the higher dosage of chemical fertilizer. Although, the amount of available P ($> 45$ mg kg$^{-1}$) and exchangeable K ($> 120$ mg kg$^{-1}$) tended to be very high in the soil treated with chicken manure, we did not observe any distinct negative effects on white mugwort growth in the current experiment. The results agreed with the recent meta-analysis showing that manure application increased available P and K [30].

### Plant growth and water content

The results showed significant differences in the fresh yield among the treatments at all three harvest times. Comparison between the control and fertilizer treatments results revealed that both organic and chemical fertilizers significantly increased the yield of the white mugwort compared to the control from the second harvest. In first and second harvests, the application of 10.08 t ha$^{-1}$ chicken manure resulted in the significantly highest fresh yield compared to the other treatments. While in the third harvest, the white mugwort yield from the treatments of 6.72 t ha$^{-1}$ and 10.08 t ha$^{-1}$ chicken manure were significantly higher than that of the other treatments (Table 4). This indicated that the high-rate application of chicken manure throughout the planting time was unnecessary. However, the effect of chemical fertilizer on the white mugwort yield was not similar to the chicken manure. In addition, the significantly highest of

**Table 3. Selected chemical properties of the soils before and after planting under the different fertilizer treatments.**

| Treatment | Soil pH (1:1 $H_2O$) | EC (1:5 $H_2O$, dS $m^{-1}$) | Organic matter[a] (g $kg^{-1}$) | Total N[b] (g $kg^{-1}$) | Available P[c] (mg $kg^{-1}$) | Available K[d] (mg $kg^{-1}$) |
|---|---|---|---|---|---|---|
| Soil before planting | | | | | | |
| | 4.10 | 0.06 | 14.00 | 0.70 | 183.00 | 77.00 |
| Soil after planting at first harvest (42 days after planting) | | | | | | |
| Control | 4.70 | 0.05 | 12.00 | 0.60 | 155.00 | 53.00 |
| CF 187.50 kg $ha^{-1}$ | 4.40 | 0.04 | 14.10 | 0.70 | 192.00 | 54.00 |
| CF 375.00 kg $ha^{-1}$ | 4.20 | 0.05 | 12.70 | 0.60 | 131.00 | 51.00 |
| CM 3.36 t $ha^{-1}$ | 4.70 | 0.05 | 14.10 | 0.70 | 148.00 | 78.00 |
| CM 6.72 t $ha^{-1}$ | 5.30 | 0.06 | 15.40 | 0.80 | 156.00 | 126.00 |
| CM 10.08 t $ha^{-1}$ | 5.10 | 0.08 | 17.20 | 0.90 | 274.00 | 210.00 |
| Soil after planting at second harvest (84 days after planting) | | | | | | |
| Control | 4.60 | 0.04 | 13.70 | 0.70 | 116.00 | 52.00 |
| CF 187.50 kg $ha^{-1}$ | 4.50 | 0.03 | 13.80 | 0.70 | 181.00 | 54.00 |
| CF 375.00 kg $ha^{-1}$ | 4.30 | 0.04 | 13.20 | 0.70 | 151.00 | 50.00 |
| CM 3.36 t $ha^{-1}$ | 4.80 | 0.04 | 14.50 | 0.70 | 122.00 | 80.00 |
| CM 6.72 t $ha^{-1}$ | 5.00 | 0.05 | 16.10 | 0.80 | 180.00 | 126.00 |
| CM 10.08 t $ha^{-1}$ | 5.10 | 0.07 | 17.30 | 0.90 | 233.00 | 182.00 |
| Soil after planting at third harvest (126 days after planting) | | | | | | |
| Control | 7.30 | 0.04 | 13.20 | 0.70 | 158.00 | 63.00 |
| CF 187.50 kg $ha^{-1}$ | 7.10 | 0.04 | 11.80 | 0.60 | 171.00 | 48.00 |
| CF 375.00 kg $ha^{-1}$ | 6.90 | 0.04 | 13.40 | 0.70 | 150.00 | 51.00 |
| CM 3.36 t $ha^{-1}$ | 6.80 | 0.07 | 15.00 | 0.80 | 171.00 | 96.00 |
| CM 6.72 t $ha^{-1}$ | 6.70 | 0.06 | 18.60 | 0.90 | 184.00 | 144.00 |
| CM 10.08 t $ha^{-1}$ | 6.80 | 0.07 | 19.50 | 1.00 | 269.00 | 228.00 |

[a] Walkley and Black method

[b] Kjeldahl method

[c] Bray II method

[d] $NH_4OAc$ method.

**Table 4. Fresh yield of white mugwort under the different fertilizer treatments.**

| Treatment | Fresh yield (t $ha^{-1}$) | | | |
|---|---|---|---|---|
| | First harvest (42 days after planting) | Second harvest (84 days after planting) | Third harvest (126 days after planting) | Total fresh yield from three harvesting |
| Control | 0.75 ± 0.04 c | 1.72 ± 0.08 e | 1.83 ± 0.18 c | 4.30 ± 0.26 d |
| CF 187.50 kg $ha^{-1}$ | 1.03 ± 0.10 b | 2.39 ± 0.08 d | 2.89 ± 0.29 b | 6.31 ± 1.06 c |
| CF 375.00 kg $ha^{-1}$ | 1.10 ± 0.08 b | 2.98 ± 0.13 c | 2.95 ± 0.12 b | 7.04 ± 0.66 bc |
| CM 3.36 t $ha^{-1}$ | 0.69 ± 0.07 c | 2.26 ± 0.17 d | 2.97 ± 0.12 b | 5.91 ± 1.04 c |
| CM 6.72 t $ha^{-1}$ | 0.82 ± 0.06 c | 3.34 ± 0.18 b | 3.51 ± 0.22 a | 7.67 ± 0.60 b |
| CM 10.08 t $ha^{-1}$ | 1.38 ± 0.03 a | 4.64 ± 0.11 a | 3.83 ± 0.23 a | 9.85 ± 0.13 a |
| F-test | ** | ** | ** | ** |
| %CV | 16.33 | 12.98 | 15.23 | 10.46 |

Data represent mean values ± standard deviation (SD). Mean with different letters in the same column indicate a significant difference according to Duncan's multiple range test at $p < 0.05$.

** Significant at $p < 0.01$.

**Table 5. Water content in the aboveground (leave and stem) of white mugwort under the different fertilizer treatments.**

| Treatment | Water content (%) | | |
|---|---|---|---|
| | First harvest (42 days after planting) | Second harvest (84 days after planting) | Third harvest (126 days after planting) |
| Control | 88.94 ± 0.71 | 88.87 ± 0.29[b] | 88.27 ± 0.53[c] |
| CF 187.50 kg ha[-1] | 89.99 ± 2.11 | 89.35 ± 0.24[ab] | 89.06 ± 0.70[abc] |
| CF 375.00 kg ha[-1] | 86.99 ± 2.18 | 89.91 ± 0.56[ab] | 90.19 ± 0.85[a] |
| CM 3.36 t ha[-1] | 85.37 ± 2.85 | 89.26 ± 1.25[ab] | 88.54 ± 0.16[bc] |
| CM 6.72 t ha[-1] | 86.13 ± 3.10 | 89.77 ± 0.06[ab] | 89.68 ± 1.01[ab] |
| CM 10.08 t ha[-1] | 87.22 ± 2.66 | 90.39 ± 0.88[a] | 90.31 ± 0.28[a] |
| F-test | ns | * | * |
| %CV | 2.74 | 0.76 | 0.74 |

Data represent mean values ± standard deviation (SD). Mean with different letters in the same column indicate a significant difference according to Duncan's multiple range test at $p < 0.05$.

ns,* Non-significant or significant at $p < 0.05$, respectively.

total fresh yield of white mugwort (9.85 t ha[-1]) from the three harvest times was obtained under the treatment of 10.08 t ha[-1] of chicken manure. The increase in the white mugwort yield may be due to the increased availability of P, K, and also the higher organic matter content after chicken manure application, as previously studied [30]. Besides the effect of macronutrients in the soil applied with chicken manure, the increased yield of the white mugwort may also be due to the higher level of micronutrients available in the chicken manure. Since chicken manure was reported to have a high concentration of macronutrients and appreciable quantities of micronutrients, such as Fe, Mn, Cu and Zn [31]. Considering the fact that some cellular metabolisms need micronutrient to function appropriately, this could explain why the organic fertilizer containing micronutrients was attributed to higher growth and yield. Similar advantages of organic fertilizer, in terms of plant growth and nutrient uptake, were reported by Hossain and Ryu [32], who suggested that organic fertilizer improved the soil physical (water holding capacity), chemical (pH, mineralization of nutrients), and biological (microbial activity) properties reflected by improved plant growth and yield. Our previous study also reported the highest net profit from white mugwort when applied at 10.08 t ha[-1] of chicken manure compared to other treatments [33].

The water content in the above-ground (leaves and stem) of the white mugwort at all harvest times are summarized in Table 5. In the first harvest, the water content in the white mugwort had no significant differences among the treatments, it ranged from 85 to 90%. However, the water content was found to be significantly higher in second and third harvests under the application of chicken manure at 10.08 t ha[-1] when compared to the control. In addition, the water content in the white mugwort rose as nitrogen levels increased. The present results agreed with the previous report by Rezaei and Pazoki [34]. The application of high N concentration improves the N uptake in plant and is employed to form soluble nitrogenous compounds, for example free amino acids such as proline, which function as osmotic regulators and take part in reducing the osmotic potential and so maintain cell turgidity [35].

## Chlorophyll content

In first and second harvests, the chlorophyll a+b content in the white mugwort leaves were not significantly altered under the different fertilizer treatments, ranging from 14.42 to 6.72 µg cm$^{-2}$ and 19.07 to 20.67 µg cm$^{-2}$, respectively. In the third harvest, however, the chlorophyll a+b content of the white mugwort manured with 10.08 t ha[-1] of chicken fertilizer increased

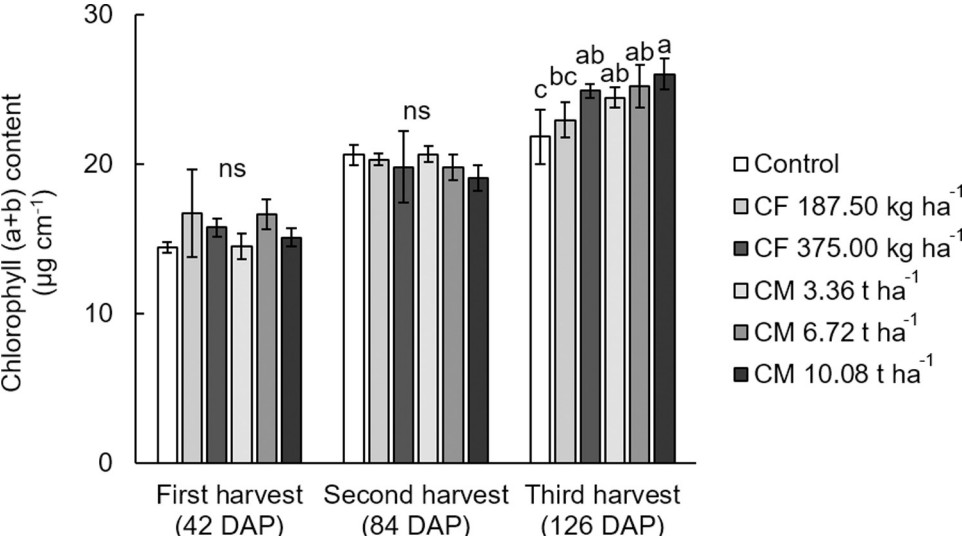

**Fig 1. Chlorophyll (a+b) content (µg cm$^{-2}$) in the above-ground (leave and stem) of white mugwort under the fertilizer treatments.** The different letters above the bars indicate statistically significant differences by Duncan's multiple range tests (p < 0.05) within each harvest. CF = chemical fertilizer (complete fertilizer 25-7-7 grade + urea 46-0-0 grade); CM = chicken manure.

significantly compared with the control and the treatment of 187.50 kg ha$^{-1}$ of chemical fertilizer (Fig 1). This may be attributed to the effect of a higher chicken manure application increases the chlorophyll content, which positively affects photosynthesis and exhibits a higher of yield [15, 36].

## Yield quality (nitrate accumulation and phytochemical compounds)

The soil with 375.00 kg ha$^{-1}$ of chemical fertilizer recorded the highest nitrate uptake in the white mugwort shoots at all three harvests, which were 23.09, 15.48, and 12.10 mg $NO_3^-$ kg$^{-1}$ fresh weight, respectively (Fig 2). The nitrate level was approximately 1–3 times higher in the chemical fertilization than in the organic fertilization. As nitrogen level increased, the plants accumulated more nitrate in the above-ground parts. Currently, there is no standard limit set for nitrate content in fresh vegetables in Thailand, however, the European Union defines the maximal nitrate level of fresh spinach and lettuce, under Commission Regulation (EC) No.1881 / 2006, as between 2,000–5,000 mg $NO_3^-$ kg$^{-1}$ in fresh weight [37]. In this study, the level of nitrate accumulation for all the white mugwort samples was 2.68–23.09 mg $NO_3^-$ kg$^{-1}$ fresh weight, which did not exceed the limit set by the EU. This is possibly due to exposure to the outdoor environment (under full sunlight condition), that even with chemical fertilizer application, the outcome showed that nitrate levels at all three harvest times of the white mugwort remained within the EC safety limit for consumers.

The levels of total phenolic content in the second harvesting plants significantly decreased with the higher dosage of both fertilizer types, whereas the dosages of the fertilizer did not significantly affect the total phenolic content in the first and third harvest plants (Fig 3). Several studies show the influence of higher fertilization, especially nitrogen fertilizer sources, on the content of phenolic compounds in plant, such as lettuce [25] and sesame [38]. Being possibly accredited to the elements necessary for phenolic synthesis, for example phenylalanine, which flows preferentially into chain protein synthesis in preference to the phenolic synthesis pathway, so increasing plant biomass in response to high nitrogen levels [39]. However, several

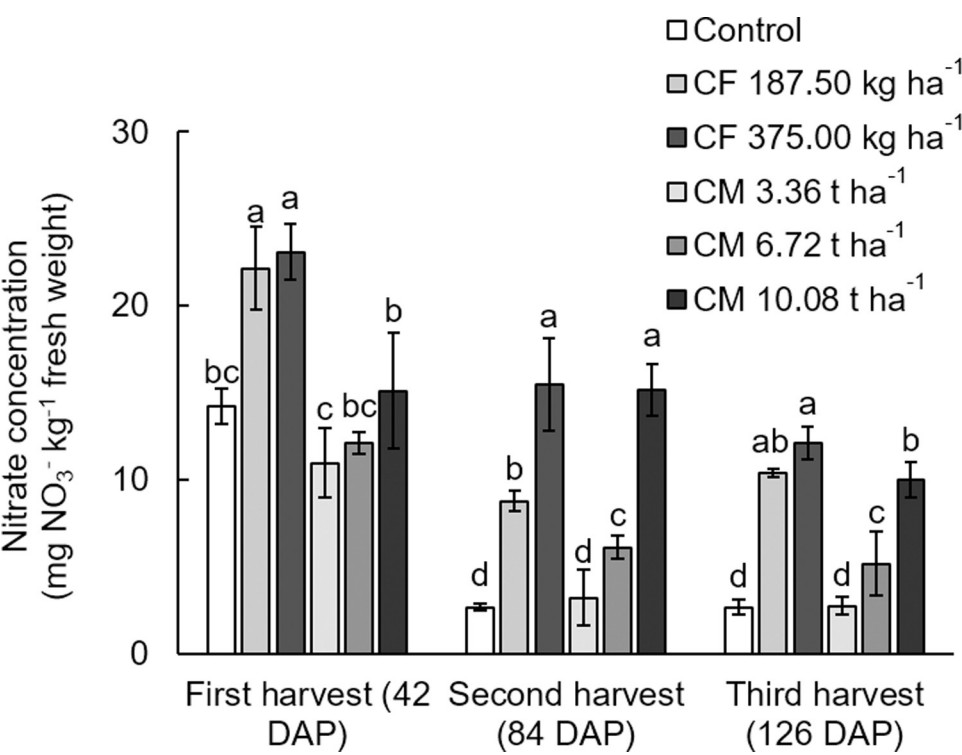

**Fig 2. Nitrate concentration (mg NO3- kg$^{-1}$ fresh weight) in the above-ground (leave and stem) of white mugwort under the fertilizer treatments.** The different letters above the bars indicate statistically significant differences by Duncan's multiple range tests (p < 0.05) within each harvest. CF = chemical fertilizer (complete fertilizer 25-7-7 grade + urea 46-0-0 grade); CM = chicken manure.

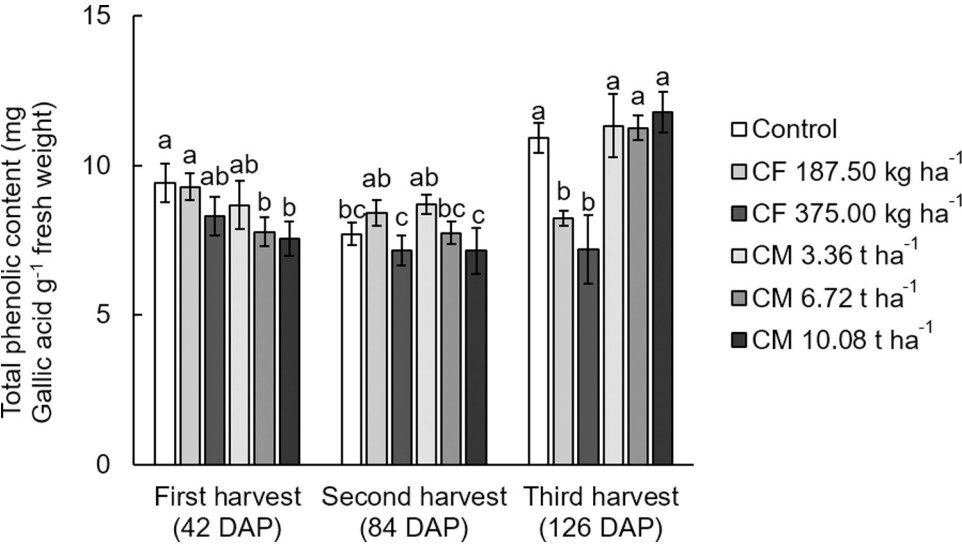

**Fig 3. Total phenolic content (mg Gallic acid g$^{-1}$ fresh weight) in the above-ground (leave and stem) of white mugwort under the fertilizer treatments.** The different letters above the bars indicate statistically significant differences by Duncan's multiple range tests (p < 0.05) within each harvest. CF = chemical fertilizer (complete fertilizer 25-7-7 grade + urea 46-0-0 grade); CM = chicken manure.

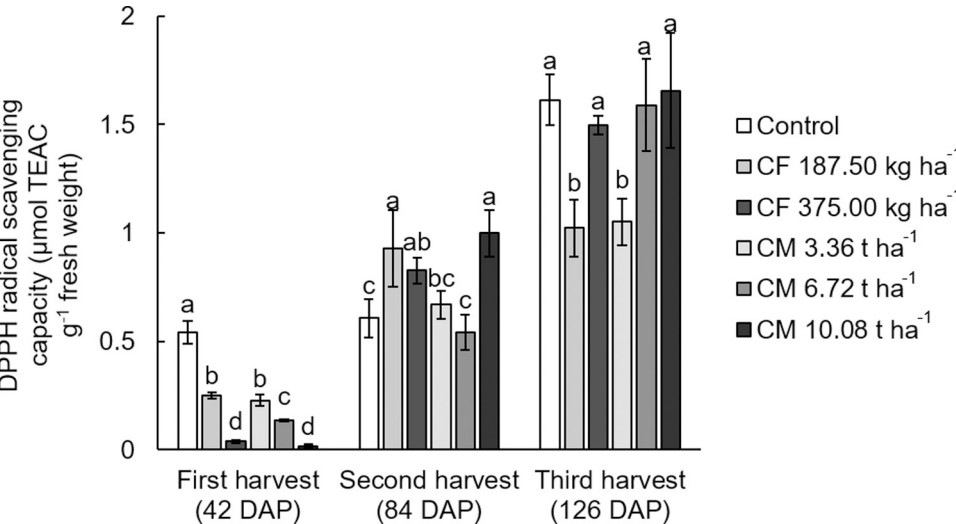

**Fig 4. DPPH radical scavenging capacity (μmol TEAC g$^{-1}$ fresh weight) in the aboveground (leave and stem) of white mugwort under the fertilizer treatments.** The different letters above the bars indicate statistically significant differences by Duncan's multiple range tests ($p < 0.05$) within each harvest. CF = chemical fertilizer (complete fertilizer 25-7-7 grade + urea 46-0-0 grade); CM = chicken manure.

investigations show an undefined trend in the total phenolic content in relation to increasing fertilization [40, 41]. Other studies also show that organic and chemical fertilizers did not affect the phenolic compounds, compared with non-fertilizer treated plant [42].

The present result showed that chemical fertilizer treated plants' total phenolic content decreased, and DPPH radical scavenging activity increased with the longer harvesting time. While chicken manure treated plants had higher total phenolic content and DPPH radical scavenging activity during the longer harvesting time. At the third harvest, the white mugwort treated all three chicken manure dosages had the highest total phenolic content (11.26–11.77 mg gallic acid equivalents g$^{-1}$ fresh weight), while the white mugwort treated with chicken manure at medium and high dosage rates showed the highest DPPH radical scavenging activity (1.59–1.65 μmol TEAC g$^{-1}$ fresh weight) (Figs 3 and 4). According to the report by Ghimire et al. [43], there is a positive correlation between micronutrients including Zn and Mn in plant tissues and the phenolic compounds of *Atractylodes japonica* Koidz. The higher total phenolic content in chicken manure treated plants with the longer harvesting time in this study may be explained by the greater opportunity for chicken manure micronutrient take-up, as it was applied after each harvesting.

Although several studies reported that more fertilization led to a reduction of DPPH radical scavenging activity [38, 44], there were no specific relationships between total phenolic content or DPPH radical scavenging capacity and the fertilizer or application level in this present study. This suggested that other factors may play a role in total phenolic content and DPPH radical scavenging activity of the white mugwort. Several studies show a correlation between total phenolic content and DPPH radical scavenging activity under other environmental conditions, such as precipitation, humidity, and solar radiation intensity [43, 45]. To support this, Kałużewicz et al. [46] revealed that the concentration of phenolic content in cauliflower is significantly increased with the amount of irrigation water. Since the average rainfall in our research area during the third harvesting cycle (350 mm) was higher than that of the first and second harvesting cycles, 230 and 250 mm, respectively [47]. The increase of total phenolic content and DPPH radical scavenging activity of the white mugwort in the third harvest may

be related to the higher rainwater during the harvest time. Previous studies have revealed a significant positive correlation between phenolic compounds and antioxidant activity [41]. However, Ibrahim et al. [6] observed that DPPH radical scavenging activity in the leaf tissues of *Labisia pumila* Benth correlated not only with total phenolic levels but also with other anti-oxidative compounds, such as ascorbic acid and glutathione. Nevertheless, the phytochemical content to environmental conditions correlation continues to be contentious. Future work should investigate the correlation between the phytochemical content to environmental conditions in greater detail.

## Conclusion

The application of organic and chemical fertilizers positively influenced the growth and yields of white mugwort grown under extremely acidic soil with scarce nitrogen. Therefore, application of nitrogen fertilizer sources could overcome this limiting factor. Chicken manure applied at 10.08 t ha$^{-1}$ produced the best white mugwort growth and yields. For a favorable outcome, higher total phenolic content and lower nitrate level was also found in the white mugwort treated with 10.08 t ha$^{-1}$ of chicken manure. This was probably due to improved soil organic matter and a sustainable supply of nutrients from the manure.

## Acknowledgments

The authors would like to Thammasat University Center of Excellence in Agriculture Innovation Center through Supply Chain and Value Chain and Major of Agricultural Technology, Faculty of Science and Technology, Thammasat University for providing experimental and laboratory facility.

## Author Contributions

**Conceptualization:** Ornprapa Thepsilvisut.

**Data curation:** Ornprapa Thepsilvisut.

**Formal analysis:** Ornprapa Thepsilvisut.

**Funding acquisition:** Ornprapa Thepsilvisut.

**Investigation:** Ornprapa Thepsilvisut.

**Methodology:** Ornprapa Thepsilvisut.

**Project administration:** Ornprapa Thepsilvisut, Sudathip Sae-Tan.

**Resources:** Ornprapa Thepsilvisut.

**Software:** Ornprapa Thepsilvisut.

**Validation:** Preuk Chutimanukul, Sudathip Sae-Tan.

**Writing – original draft:** Ornprapa Thepsilvisut.

**Writing – review & editing:** Ornprapa Thepsilvisut, Preuk Chutimanukul, Sudathip Sae-Tan, Hiroshi Ehara.

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
