## [Decision Letter · Decision Letter 0]

10 Jan 2022

PONE-D-21-36764Effect of chicken manure and chemical fertilizer on the yield and qualities of white mugwort at different harvesting timesPLOS ONE

Dear Dr. thepsilvisut,

Thank you for submitting your manuscript to PLOS ONE. After careful consideration, we feel that it has merit but does not fully meet PLOS ONE’s publication criteria as it currently stands. Therefore, we invite you to submit a revised version of the manuscript that addresses the points raised during the review process. Please submit your revised manuscript by Feb 24 2022 11:59PM. If you will need more time than this to complete your revisions, please reply to this message or contact the journal office at plosone@plos.org. Please include the following items when submitting your revised manuscript:A rebuttal letter that responds to each point raised by the academic editor and reviewer(s). You should upload this letter as a separate file labeled 'Response to Reviewers'.A marked-up copy of your manuscript that highlights changes made to the original version. You should upload this as a separate file labeled 'Revised Manuscript with Track Changes'.An unmarked version of your revised paper without tracked changes. You should upload this as a separate file labeled 'Manuscript'.

We look forward to receiving your revised manuscript.

Kind regards,

Umakanta Sarker

Academic Editor

PLOS ONE

Journal Requirements:

Reviewers' comments:

Reviewer's Responses to Questions

**Comments to the Author**

1. Is the manuscript technically sound, and do the data support the conclusions?

Reviewer #1: Yes

Reviewer #2: Yes

2. Has the statistical analysis been performed appropriately and rigorously? 

Reviewer #1: Yes

Reviewer #2: Yes

3. Have the authors made all data underlying the findings in their manuscript fully available?

Reviewer #1: Yes

Reviewer #2: Yes

4. Is the manuscript presented in an intelligible fashion and written in standard English?

Reviewer #1: Yes

Reviewer #2: Yes

5. Review Comments to the Author

Reviewer #1: Comments and suggestions for author:

Overall comments:

The authors have prepared a nice and attractive research article. This article contains new findings regarding the application of organic and inorganic fertilizers for the cultivation of the medicinal herb 'white mugwort'. The data presentation and statistical analysis are good. The author needs to improve the English language and writing style of this article. The author arranged and presented all the data in the table. The author must arrange the data in both table and graph. Actually, data on chlorophyll content, total phenolic content, nitrate reductase activity, and DPPH radical scavenging capacity need to present in a graph with a standard error bar. In the introduction chapter, the author needs to add more examples of the application of organic and chemical fertilizers for the cultivation of white mugwort plants, not for other vegetables crops or plants. If the author would arrange a treatment with the combination of both chemical and organic fertilizers this article could be more reasonable and scientifically sound.

Specific Comments:

1. Total Plagiarism percentage is 36 % with the yellow color level in Turnitin that is not acceptable to my knowledge. The author should reduce the plagiarism from 36% to 20% with blue color index in Turnitin software. In line 28-32, In line 103-104, Line 147-150, Line 160, Line 164-176, Line 179-182, Line 246-248, Line 287-288, Line 325-327 plagiarism percentage is not acceptable. Please reduce it.

2. Line 116: What types of natural extract were applied in the plot to control pests and diseases?

3. Line 112-115: Please "The organic fertilizer treatments with chicken manure were applied at the time of 113 transplantation, whereas the chemical fertilizer treatments were applied with complete fertilizer 114 25-7-7 grade at the time of transplantation, and with urea 46-0-0 grade at 14 days before harvest." change into " The doses of organic fertilizers "chicken manure" were applied at the time of transplanting, whereas the doses of chemical fertilizers were applied with complete fertilizer 114 25-7-7 grade at the time of transplanting, and with urea 46-0-0 grade at 14 days before harvest.

4. Please provide the analyzed data in SPSS software with a PDF file for reviewer understanding.

Reviewer #2: Although the manuscript was written in standard English, statistical analysis were performed properly, I found the concept of this manuscript very common and weak particularly to be published in this journal. I believe authors can improve this manuscripts by either improving the presentation and/or present additional critical data.

6. PLOS authors have the option to publish the peer review history of their article (what does this mean?). If published, this will include your full peer review and any attached files.

Reviewer #1: No

Reviewer #2: No

---

## [Author Response · Author response to Decision Letter 0]

23 Feb 2022

We are most grateful to you and the reviewers for helpful comments on the original version of our manuscript and consider our paper for publication in the journal. Follow the reviewer's comments we have revised the manuscript.

We deeply hope that this revised version of our manuscript in now suitable for publication.

Thank you very much indeed.

Sincerely yours,

Assist. Prof. Dr. ORNPRAPA THEPSILVISUT

Major of Agricultural Technology

Faculty of Science and Technology

Thammasat University

Klong Noeng, Klong Luang, Pathum Thani 12120

Tel: +66(0)2-564-4440 Ext 2356

Mobile: +66(0)96 995 3542

E-mail: ornprapa@hotmail.com, ornprapa@tu.ac.th

---

## [Decision Letter · Decision Letter 1]

16 Mar 2022

Effect of chicken manure and chemical fertilizer on the yield and qualities of white mugwort at dissimilar harvesting times

PONE-D-21-36764R1

Dear Dr. thepsilvisut,

We’re pleased to inform you that your manuscript has been judged scientifically suitable for publication and will be formally accepted for publication once it meets all outstanding technical requirements.

**During proof reading:**

Add spaces before and after the symbol “˂”, and “±”. Follow this style throughout the whole MS where it exists.

Table 2: in column 1: unify the font of all writing.

Change small letter “x” to the symbol of the cross “×”. Follow this style throughout the whole MS where it exists.

Kind regards,

Umakanta Sarker

Academic Editor

PLOS ONE

Additional Editor Comments (optional):

During proof reading:

Add spaces before and after the symbol “˂”, and “±”. Follow this style throughout the whole MS where it exists.

Table 2: in column 1: unify the font of all writing.

Change small letter “x” to the symbol of the cross “×”. Follow this style throughout the whole MS where it exists.

Reviewers' comments:

Reviewer's Responses to Questions

**Comments to the Author**

1. If the authors have adequately addressed your comments raised in a previous round of review and you feel that this manuscript is now acceptable for publication, you may indicate that here to bypass the “Comments to the Author” section, enter your conflict of interest statement in the “Confidential to Editor” section, and submit your "Accept" recommendation.

Reviewer #1: All comments have been addressed

Reviewer #2: All comments have been addressed

2. Is the manuscript technically sound, and do the data support the conclusions?

Reviewer #1: Yes

Reviewer #2: Yes

3. Has the statistical analysis been performed appropriately and rigorously? 

Reviewer #1: Yes

Reviewer #2: Yes

4. Have the authors made all data underlying the findings in their manuscript fully available?

Reviewer #1: Yes

Reviewer #2: Yes

5. Is the manuscript presented in an intelligible fashion and written in standard English?

Reviewer #1: Yes

Reviewer #2: Yes

6. Review Comments to the Author

Reviewer #1: Dear author,

Thank you for submitting the revised version of the manuscript. The revisions of the manuscript were made accordingly to reviewer comments.

Reviewer #2: Although such findings are most times necessary to be confirmed by conducting the experiment at least two times, i believe this manuscript will set the stage for more rigorous research in the area of white mugwort fertilization.

Best Regards.

7. PLOS authors have the option to publish the peer review history of their article (what does this mean?). If published, this will include your full peer review and any attached files.

Reviewer #1: **Yes: **Haque Md Azadul

Reviewer #2: No

---

## [Editor Report · Acceptance letter]

25 Mar 2022

PONE-D-21-36764R1 

Effect of chicken manure and chemical fertilizer on the yield and qualities of white mugwort at dissimilar harvesting times 

Dear Dr. Thepsilvisut:

I'm pleased to inform you that your manuscript has been deemed suitable for publication in PLOS ONE. Congratulations! Your manuscript is now with our production department. 

Kind regards, 

on behalf of

Professor Umakanta Sarker 

Academic Editor

PLOS ONE